# Molecule Diffusion Behavior of Tritium and Selenium in Mongolia Clay Rock by Numerical Analysis of the Spatial and Temporal Variation

**Chuan-Pin Lee** [1,†], **Yanqin Hu** [1,†], **Neng-Chuan Tien** [2,*], **Shih-Chin Tsai** [2,*], **Yunfeng Shi** [1,3], **Weigang Liu** [1], **Jie Kong** [1] **and Yuzhen Sun** [1]

1   School of Nuclear Science and Engineering, East China University of Technology, Nanchang 330013, China; bennis6723@139.com (C.-P.L.); hyqdrx@163.com (Y.H.); 541006935@126.com (Y.S.); lwg0002021@163.com (W.L.); kongjie127@163.com (J.K.); sunyuzhen1210@126.com (Y.S.)
2   Nuclear Science and Technology Development Center, National Tsing Hua University, Hsinchu 30013, Taiwan
3   Department of Nuclear Environmental Science, China Institute for Radiation Protection (CIRP), Taiyuan 030006, China
*   Correspondence: nctien@mx.nthu.edu.tw (N.-C.T.); sctsai@mx.nthu.edu.tw (S.-C.T.); Tel.: +86-1825824-4042 (N.-C.T.)
†   These authors contributed equally to this work.

**Abstract:** Clay rock in the Tamusu (TMS) area in the Inner Mongolia province is one candidate as a geological disposal site for highly radioactive waste in China. The diffusion behavior of HTO and Se(IV) in Tamusu clay rock was studied by through-diffusion (TD) experiments and numerical analysis to determine the spatial and temporal variation. A minimum error analysis was conducted to determine the HTO and Se(IV) diffusion coefficients in compacted TMS clay. The TD experimental results and numerical analysis showed that the diffusion of HTO and Se reached a steady state within 7 and 50 days, respectively, and the apparent diffusion coefficients ($D_a$) decreased with the increases in the compacted density. In fact, there was retardation of Se diffusion in intact TMS clay rock. A two-site sorption model for Se was applied to simulate fast and slow sorption behavior quantitatively.

**Keywords:** TMS clay rock; numerical analysis; HTO; Se; diffusion coefficients; through-diffusion (TD)

## 1. Introduction

The deep geological disposal of radioactive waste has received much attention mainly in recent years due to its safe and stable properties for radioactive waste treatment by many nuclear countries. The "multibarrier" concept has been applied to deal with issues related to radioactive waste repositories, such as the use of clay minerals as an engineering (buffer/backfill) material because of their mechanical stability and better sorption characteristics. Furthermore, firm and stable sedimentary formations are highly suitable as host rocks in the final disposal repository [1].

The retention of spent-fuel radionuclides in solid clay rocks controls the relatively low velocity of their transport to the geological environment. There are several suitable properties for the retention of radionuclides, such as low hydraulic conductivity, strong sorption behavior of radionuclides, and the slow migration caused mainly by molecular diffusion, and all of these advantageous properties make these rocks acceptable candidates for hosting radioactive waste [2].

Hence, it is necessary to discuss and clarify the retardation capacity of radionuclides in host rock in radioactive waste repositories. In terms of disposal materials, igneous rocks (granite) and sedimentary rocks (clay formations) are generally selected as the major host media for disposal repositories [3]. For instance, the Olkiluoto discrete granitic island (Finland), the Oligocene Boom Clay (Belgium), and Callovo-Oxfordian argillites (France) have each been chosen as kinds of materials for disposal repositories in Belgium and

in the eastern part of the Paris Basin, France. Additionally, the Opalinus Clay (OPA) in Switzerland is also a potential host rock [4,5].

Previously, several types of diffusion studies were performed on Beishan granite (in the northwest of China) for the disposal of high-level radioactive waste (HLW) in the 1980s [6–10]. The existence of fractures in the granite can result in faster radionuclide migration when groundwater enters the repository.

Recently, clay (sedimentary) rock in the Tamusu (TMS) area was considered as a potential geological disposal site during safety assessments for highly radioactive waste in China [11,12]. Among several other advantages, the permanent negative charge on the clay surface and its low permeability make the clay an ideal repository candidate. Furthermore, clay (sedimentary) rock shows high adsorption for cations in groundwater, but it simultaneously strengthens the anion exclusion effect between negative radionuclides (anions) and the permanent negative charge on the clay surface [13–15].

As one of the primary fission products, selenium-79 shows a long half-time ($t_{1/2} = 3.56 \times 10^5$ a) and high radioactivity, and it poses the main potential risks from HLW disposal repositories [16]. Effectively, selenium speciation, which occurs in various valence states (selenium (0), selenide (-II), selenite (IV), and selenate (VI)), depends on various pH, dissolved oxygen (DO), and redox potential conditions in solution. In fact, the mobility of selenate (VI) and selenite (IV) in groundwater is obviously much better than that of the others (selenium (0) and selenide (-II)). Therefore, according to a safety assessment for evaluating radionuclide release to the environment from an HLW repository, faster transport of selenate (Se(VI)) and selenite (Se(IV)) from HLW repositories to the geological environment is considered a major issue and concern. Several previous studies have been performed to obtain the sorption and diffusion parameters, $K_d$ values, and the diffusion coefficients under various conditions [17,18]. In those previous works, the apparent diffusion coefficient values obtained from through-diffusion experiments in crushed mudrock and granite were two orders of magnitude different: $1.04 \times 10^{-12}$ and $1.40 \times 10^{-10}$ m$^2$ s$^{-1}$ in synthetic groundwater, respectively. In fact, the distribution coefficient ($K_d$) for selenite (Se(IV)) was determined with the sorption of various Se species on a mineral surface, and the results basically suggested that the $K_d$ values of selenite (Se(IV)) on the mudrock appeared to increase with the decrease in Se(IV) equilibrium concentration in the solution.

This is expected to be representative of all radionuclides due to the fact that tritiated water (HTO) shows no retention in the solid phase. Additionally, two important parameters, effective porosity and retardation factor, only related to the compacted samples could be derived from HTO through-diffusion (TD) experiments and a novel analytical method by numerical simulation [19]. In this paper, through-diffusion of a nonreactive radiotracer (HTO) in compacted TMS clay and Se diffusion experiments in intact TMS samples were applied to investigate the retention behavior based on the HTO and Se(IV) diffusion coefficients. Moreover, the concentration profiles with the influence of the spatial and temporal variability of HTO and Se(IV) were investigated and analyzed using a numerical method for steady-state prediction. Furthermore, a two-site sorption model for Se(IV) was applied to simulate fast and slow sorption behavior quantitatively to realize the retention of Se in TMS clay rocks. Finally, the experimental and numerical results of HTO and Se(IV) in this work may serve as an important reference case for future safety assessments of clay rock repositories in China.

## 2. Materials and Methods

### 2.1. Theory of Through-Diffusion

In this work, Fick's second law was applied to obtain the diffusion coefficient (D) for quantitative analysis of the transport of radionuclides in compacted media. This equation can be expressed as follows:

$$\frac{\partial c}{\partial t} = D \left( \frac{\partial^2 C}{\partial x^2} + \frac{\partial^2 C}{\partial y^2} + \frac{\partial^2 C}{\partial z^2} \right) \tag{1}$$

The diffusion coefficient D is then determined by the molecule spread rate, which is the proportionality constant between the mass flux and the solute concentration gradient; in one dimension, the diffusion equation for the radionuclides can be written as follows:

$$\frac{\partial c}{\partial t} = D_a \frac{\partial^2 C}{\partial x^2}, \ \left( D_a = \frac{D_e}{\alpha}, \alpha = \theta + \rho_b K_d \right) \tag{2}$$

where $D_a$ and $D_e$ are apparent and effective diffusion coefficients, respectively; $\alpha$ depends on the porosity of compacted samples ($\theta$), the bulk density of the dry material ($\rho_b$), and the distribution coefficient $K_d$; and $C$ is the solute concentration in the liquid phase. Both initial and boundary conditions limit the through-diffusion method. Each boundary condition can be expressed as follows:

$$C(x,0) = 0, \ 0 < x < LC(0,t) = C_0 \, C(L,t) \sim 0$$

where the concentration in the reservoir containing the tracer is constant ($C_0$), the concentration in the opposing reservoir is kept close to zero, and $L$ is the overall length of the compacted samples.

The analytical solution for the concentration profile $C(x,t)$ in the compacted samples at time ($t$) is given by Crank (1975) [20]:

$$\frac{C(x,t)}{C_0} = 1 - \frac{x}{L} - \frac{2}{\pi} \sum_{n=1}^{\infty} \frac{1}{n} \sin \left[ \frac{n\pi x}{L} \right] \exp \left[ \frac{-n^2 \pi^2 D_a t}{L^2} \right] \tag{3}$$

The formula of the cumulative mass ($M$) or concentration ratio (CR(t)) of tracer in the measurement reservoir is defined by [20–24]:

$$M = C_0 \alpha LS \left( \frac{D_a t}{L^2} - \frac{1}{6} - \frac{2}{\pi^2} \sum_{n=1}^{\infty} \frac{(-1)^n}{n^2} \exp \left[ \frac{-n^2 \pi^2 D_a t}{L^2} \right] \right) \tag{4}$$

$$\mathrm{CR(t)} = \frac{\sum C(t)}{C_0} = \frac{\alpha LS}{V} \left( \frac{D_a t}{L^2} - \frac{1}{6} - \frac{2}{\pi^2} \sum_{n=1}^{\infty} \frac{(-1)^n}{n^2} \exp \left[ \frac{-n^2 \pi^2 D_a t}{L^2} \right] \right) \tag{5}$$

where $M$ is the cumulative mass of tracer, $D_a$ is the apparent diffusion coefficient that takes the rock capacity factor ($\alpha$) into consideration, and $S$ is the cross-section area of the compacted samples. For a sufficient TD experimental time, the diffusion process will reach a steady state; subsequently, in Equations (3)–(5), the exponential term tends to zero, and the concentration profile curve, $C(x.t)$ (or $M$ or CR(t)), shows a linear relationship with time ($t$). Moreover, a numerical analysis was developed for the spatial and temporal variability $C(x.t)$ of HTO and Se(IV) in this study, and those figures were compared and discussed with the diffusion time for the steady state. Here, two different algorithms were implemented, namely the trust-region reflective algorithm and the Levenberg–Marquardt algorithm. Using both provides an effective and important tool in safety assessment for future clay rock repositories when based on concurrent experimental and numerical results of HTO and Se(IV).

*2.2. Experiments*

2.2.1. Clay Rocks and Liquids

TMS clay rock samples and synthetic groundwater (GW) were investigated in previous studies, including several major and minor mineral components such as quartz, feldspar, dolomite, ankerite, and wairakite, identified by X-ray diffraction spectra [11,12,25,26]. All TMS clay rocks in this work were crushed and pretreated with a grinding machine, and the clay powders with a particle size of about 1 um were passed through 200 sieves. Moreover, an intact TMS clay rock sample was cut and treated directly from a borehole in Inner Mongolia. Before testing in batch tests and through-diffusion experiments (TD), the samples were washed several times, flushed, and rinsed for 15 to 30 min with deionized

water (DIW) and placed into a $100 \pm 5\,°C$ oven for 24 to 48 h. Finally, the rock samples were stored in desiccators for the batch and column tests.

### 2.2.2. Batch Tests

ASTM batch tests were applied in this work [27]. Three sets of TMS samples were prepared in centrifugal tubes (50 mL) for the testing condition, including sorption kinetic experiments, sorption isotherm experiments, and microanalysis and elemental analyses using a scanning electron microscope (SEM, NNS-450, Philips, Amsterdam, The Netherlands) equipped with an energy dispersive spectrometer system (EDS, Oxford E-MAX, Oxford, UK). A Se(IV) stock solution ($C_0$) providing stable isotope tracers and selenium dioxide (SeO$_2$, Sigma-Aldrich, Darmstadt, Germany) was added to the GW solution prior to batch tests. All batch tests were conducted with a solid/liquid ratio of 0.04 g/20 mL.

### 2.2.3. Sorption Kinetic Experiments

A volume of 20 mL was mixed with crushed TMS clay rocks at $25 \pm 1\,°C$ for up to three days, which was found to be sufficient for attaining equilibrium. After 0.1, 0.5, 1, 2, 4, 8, 12, 24, 36, 48, and 72 h of shaking, tubes were removed and centrifuged at $10,380 \times g$ (TGL-16, XIANGYI Co., Ltd., Changsha, China) for 15 min. The pH and Eh values of the aqueous solution were also measured using a glass electrode (InoLab-412, Mettler Toledo, Switzerland) and a platinum glass electrode. The concentrations of Se(IV) versus time ($C_t$) were measured using inductively coupled plasma optical emission spectrometry (ICP-OES, iCAP 7000, Thermo, Waltham, MA, USA).

One- and two-site sorption models were applied to reach the equilibrium of the radionuclide sorption kinetic mechanism in a solid/liquid system [28]. The expression of the ratio of the Se(IV) concentration in the liquid phase ($C_t$) and the initial concentration ($C_0$) as a function of time follows:

$$C_t/C_0 = C_e + (1 - C_e)\exp\hat{}(-\lambda_1\, t) \tag{6}$$

$$C_t/C_0 = C_e + (1 - C_e)\exp\hat{}(-\lambda_1\, t) + (1 - f)(1 - C_e)\exp\hat{}(-\lambda_2\, t) \tag{7}$$

where $\lambda_1$ and $\lambda_2$ represent the one- and two-site decay constants, respectively, and $f$ is the proportionality constant between the amount of one specific site. When the reaction reached equilibrium (i.e., length of time is sufficient), the ratio of sorption $C_t/C_0$ was an exponential decay function with an equilibrium concentration $C_e$.

### 2.2.4. Sorption Isotherm Experiments

After reaching the sorption equilibrium of Se(IV) on TMS clay, a sorption isotherm experiment was conducted using initial Se(IV) concentrations ranging from $10^{-3}$ to $10^{-6}$ M. Generally, the equilibrium concentration of Se(IV) that was adsorbed on the TMS clay reached a constant and stable value with increasing Se(IV) concentrations. A Langmuir-type sorption equation consistently suggested analysis of the sorption capacity of the Se(IV) or other radionuclides in clay rocks. The Langmuir isotherm model is expressed as follows:

$$Q = \frac{MKC}{1 + KC} \tag{8}$$

where $C$ and $Q$ are the equilibrium concentrations of Se(IV) in aqueous solutions and in TMS clay, respectively. Two parameters, the bonding energy coefficient ($K$, cm$^3$/mol) and the maximum sorption ($M$, mol/g), were used to describe and understand the affinity and the sorption capacity of the TMS clay rocks, respectively.

### 2.2.5. Microanalysis and Elemental Analysis (SEM–EDS Experiments)

After pretreatment, the crushed TMS clay rocks (~1 g) were sprayed and glued into prepared centrifugal tubes (50 mL); the clay rock samples were then mixed and shaken with 40 mL of a stock solution containing Se(IV) at a concentration of $10^{-2}$ M. After 7 days, the

samples were removed and rinsed quickly with DIW to remove excess Se(IV) solution on the sample surface. After drying, the major sorption of Se(IV) on TMS clay was identified and compared using SEM–EDS with an accelerating voltage of 20 kV and a current of 10 μA. EDS was used to analyze the corresponding Se elemental composition of the clay samples.

### 2.3. Through-Diffusion Experiments (TD): Column Tests

A reliable and precise TD device with sandwich-like columns, shown in Figure 1, was developed and modified as described in previous works; the experimental apparatus has been shown in several reports [17–19,22–24] and was used in this work for the TD experiments. The apparatus consisted mainly of a highly precise peristaltic and isocratic flow pump (MASTERFLEX L/S, Cole-Parmer Instrument Co., Barrington, USA) for 5 polypropene (PP) columns (Nos. 1–5, modified type, Hsinchu ZeGi Industrial Co., Ltd., Hsinchu, Taiwan (R.O.C)) and an initial tracer reservoir, including the radiotracer (HTO) and the associated Teflon® (PTFE) units and connectors. The PP columns consisted of pressure-resistant polypropene (<10 MPa) with a length of 13.6 cm and an inner diameter of 5 cm, which was filled with crushed clay rock powders having a total porosity of 0.1 to 0.4 and a bulk density of 1.6 to 2.4 g/cm$^3$.

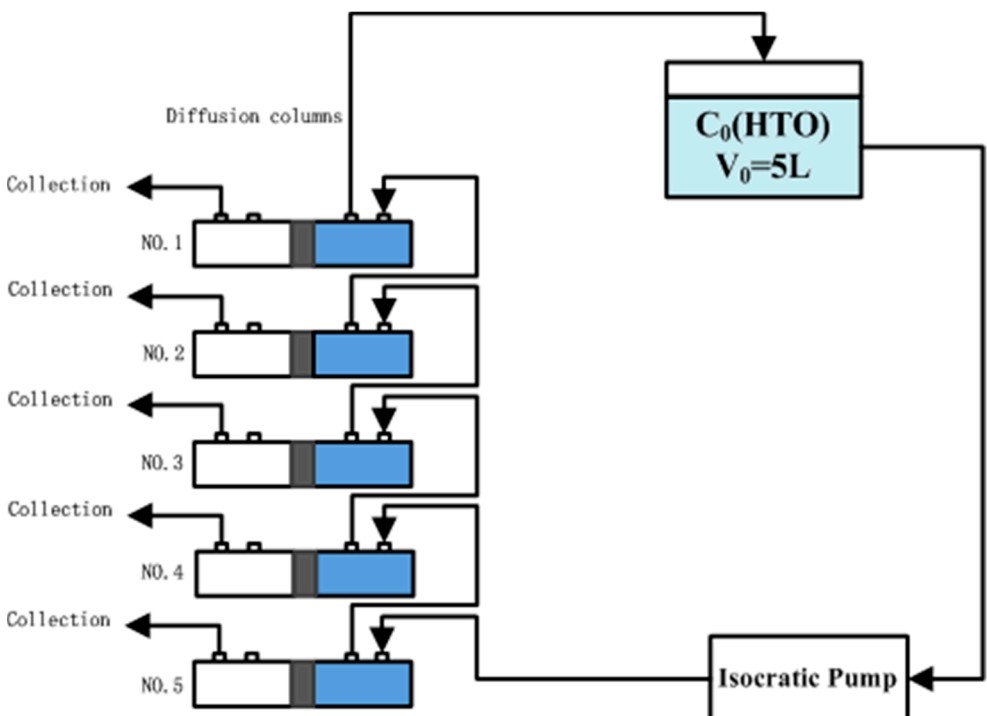

**Figure 1.** The reliable and precise TD device with a sandwich-like column apparatus used in this study [17–19,22–24].

### 2.3.1. Water Saturation

The PP columns (Nos. 1–5) for TMS clay, described in Table 1, were compacted with the same diffusion length (L = 0.3 cm) and cross-section area (diameter: 5 cm) in a bulk density of 1.6 to 2.4 g/cm$^3$. In order to make sure each pore space in the compacted TMS clay was filled and saturated with water before the TD experiments, the multichannel switch valves were only connected to the GW reservoir without the HTO radiotracer. During the water saturation period (7 to 30 days), approximately 3 to 5 mL effluent from 5 columns was sampled every 7 days. The concentrations of Na, Mg, Ca, and K versus time ($C_t$) in the effluent were measured using inductively coupled plasma optical emission spectrometry (ICP-OES, iCAP 7000, Thermo). It was evaluated and determined that water saturation had been reached once the Na, Mg, Ca, and K values were within 5% of the corresponding liquid phase concentrations.

**Table 1.** The through-diffusion conditions for the HTO and Se(IV) in compacted and intact clay rocks.

| Item. Type RN | 1 | | | | | 2 |
|---|---|---|---|---|---|---|
| | Compacted Powder | | | | | Intact Rocks |
| | HTO | | | | | Se(IV) |
| Column | No. 1 | No. 2 | No. 3 | No. 4 | No. 5 | No.6 |
| Bulk density | 1.6 | 1.8 | 2.0 | 2.2 | 2.4 | - |
| Initial HTO activity $A_0$ (dpm/mL)/Se Conc. Co (ppm) | HTO: 40 dpm/mL ($V_0$ = 5000 mL) | | | | | Se(IV): 3000 ppm |

### 2.3.2. Nonreactive Tests—HTO

A nonreactive radiotracer (HTO) was applied to determine the effective porosity of TMS clay compacted in different PP columns (Nos. 1–5), done in order to characterize the major physical transport processes in the TD column system. For plotting accumulative breakthrough curves (BTCs) versus time, Samples 1–5 in diffusion cells were periodically collected and measured by a liquid scintillation counter (LSC, Packard 3170 AB/TR, Shelton, CT, USA). Sampling and recycling were conducted in two steps. The first step was often a manual sampling step. The experimental procedure was as follows: (a) First, periodical samples were collected daily as in previous work [22–24], and one cycle was flushed by flow pump over 2 to 3 volumes in the source end of the PP columns (Nos. 1–5); the pump was then stopped and the PP columns were kept in a static condition. (b) An aliquot (5 mL) of the samples collected at different times in the diffusion end was then sampled for measuring the HTO activities by LSC. Table 1 shows that this experimental process and these conditions were set up and repeated until a steady state was reached (7–10 days).

### 2.3.3. Reactive Tests—Selenium (Se(IV))

TD experiments with various reactive radiotracers were generally effectively applied and designed for inadequate transport models in a groundwater system through a calibration/validation process [19]. After the HTO was completely finished, reactive tracers (Se) were applied to find out the sorption or anion exclusion effect on radionuclide diffusion in TMS clay that was spiked with Se(IV). Moreover, an intact TMS clay rock sample was compacted into a stainless column (0.3 cm length and 4.5 cm diameter) in previous work [29], which was also conducted in this study in order to simulate a realistic underground repository. The corresponding accumulative concentration ratio (CR(t)) of Se(IV) was analyzed and determined by ICP-OES. Table 1 also concludes a series of experimental conditions in TD column tests for Se(IV).

## 3. Results

### 3.1. Batch Sorption Tests: One- and Two-Site Kinetic Fitting and Langmuir Isotherm

During the batch tests, the pH in the liquid phase ranged from 7.5 to 7.1 at the initial and the final equilibrium of the sorption experiment. Although the pH of the GW in the batch sorption tests did not change much, it decreased with time. Figure 2a and Table 2 show the Se sorption and experimental results in GW at 0.04 g/20 mL, respectively, which show that the experimental data were applied to fit with two exponential functions. Moreover, the two-site fitting is characterized by two decay constants ($\lambda_1$, and $\lambda_2$), while there is only one decay constant for the other by fitting the normalized concentration (i.e., $C_t/C_0$). Table 2, which compares the least-squares errors (LSEs), suggests that the two-site fitting curves may be more suitable than those with only one site in describing the sorption kinetics of Se on TMS clay in GW. After a trial-and-error fitting process, a Langmuir model was found to obtain numerical results matching our experimental data at various initial Se(IV) concentrations. Figure 2b and Table 3 list the fitting parameters of the Langmuir model.

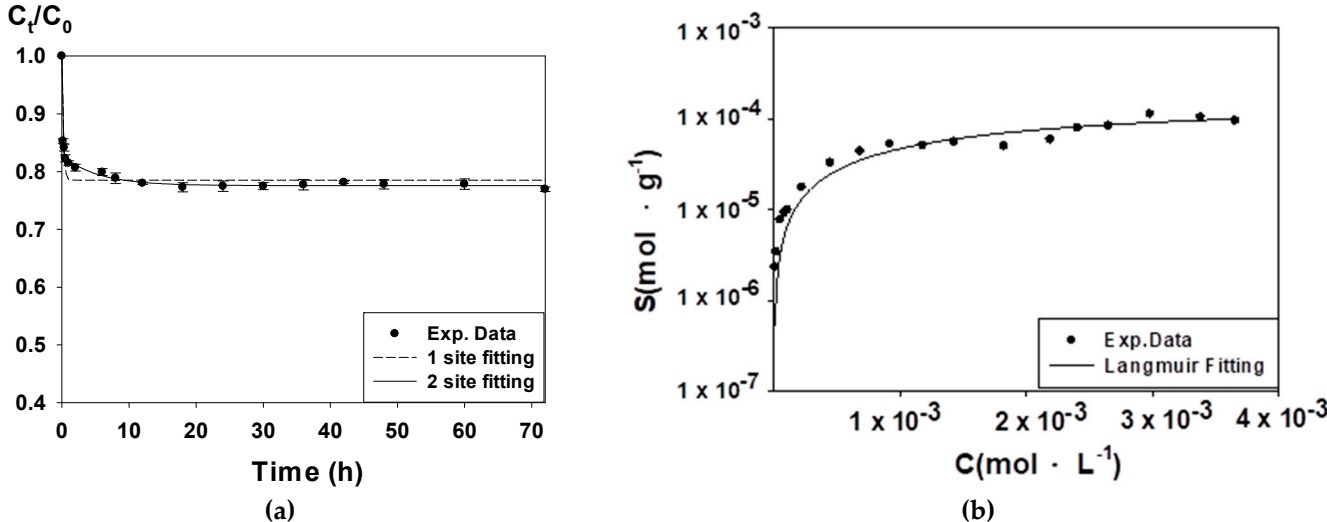

**Figure 2.** Sorption of Se on TMS clay: (**a**) kinetic sorption tests; (**b**) Langmuir isotherm.

**Table 2.** Model fitting parameters for sorption of Se on TMS clay.

| Parameters | Sorption | |
|---|---|---|
| | 1-Site | 2-Site |
| $C_e$ | $7.85 \times 10^{-1}$ | $7.76 \times 10^{-1}$ |
| λ1 | $4.88 \times 10^{0}$ | $1.02 \times 10^{1}$ |
| λ2 | - | $1.86 \times 10^{-1}$ |
| $f$ | - | $7.67 \times 10^{-1}$ |
| LSE | $3.48 \times 10^{-3}$ | $3.29 \times 10^{-4}$ |

**Table 3.** Langmuir fitting parameters for sorption of Se on TMS clay.

| Parameter | K | M (mol/g) | R-Squared |
|---|---|---|---|
| Se(IV) | $3.67 \times 10^{2}$ | $1.75 \times 10^{-4}$ | 0.9261 |

*3.2. Microanalysis and Elemental Analysis for Se Sorption on TMS Clay Rocks*

According to the SEM photos, there are layers and fine particles in TMS clay rocks, shown in Figure 3a, because these rocks occur as lithified sediment particles. Moreover, Figure 3b,c shows the corresponding images of SEM–EDS that were analyzed and compared by Se mapping analysis. The comparison of the mineral components in TMS clay rock, shown in Figure 3c (green), indicates that they are responsible for the sorption of Se(IV) [26,29,30].

*3.3. Diffusion Coefficients of HTO and Se for TMS Clay Rocks*

The diffusion parameters regressed and, as calculated using Equations (3)–(5) using the Crank method [20], are listed in Table 4 by the accumulative concentration curve's (CR(t)) linear relationship with slope and interception. It summarizes the TD experiments on various compacted densities (1.6 to 2.4 cm$^3$/g) and intact TMS clay rocks, and the accumulative concentration curves (CR(t)) of HTO and Se obtained by the TD experiments are shown in Figure 4. This indicates that the time lag between HTO and Se in TMS clay rock to diffuse out is approximately 1 and 30 days, respectively, and it reached a diffusion steady state after about 7 and 80 days due to constant diffusing flux. The dimensionless parameter $t_d = (D_a \cdot t_f / L^2)$, an important factor, is introduced here to determine if the diffusion reached equilibrium. Crank (1975) stated and suggested that diffusion steady state will be achieved when $t_d > 0.45$. The TD results showed that the td values of all columns were higher than 0.45 and good R-squared values (R$^2$ > 0.9) were obtained in five columns, and Se(IV)

exhibited obvious retardation behavior in intact TMS clay rocks. Moreover, it showed the lowest diffusion coefficients ($D_a = 1.10 \times 10^{-12}$ m$^2$/s and $D_e = 3.24 \times 10^{-12}$ m$^2$/s) in Se(IV) rather than in HTO in all columns (Table 4).

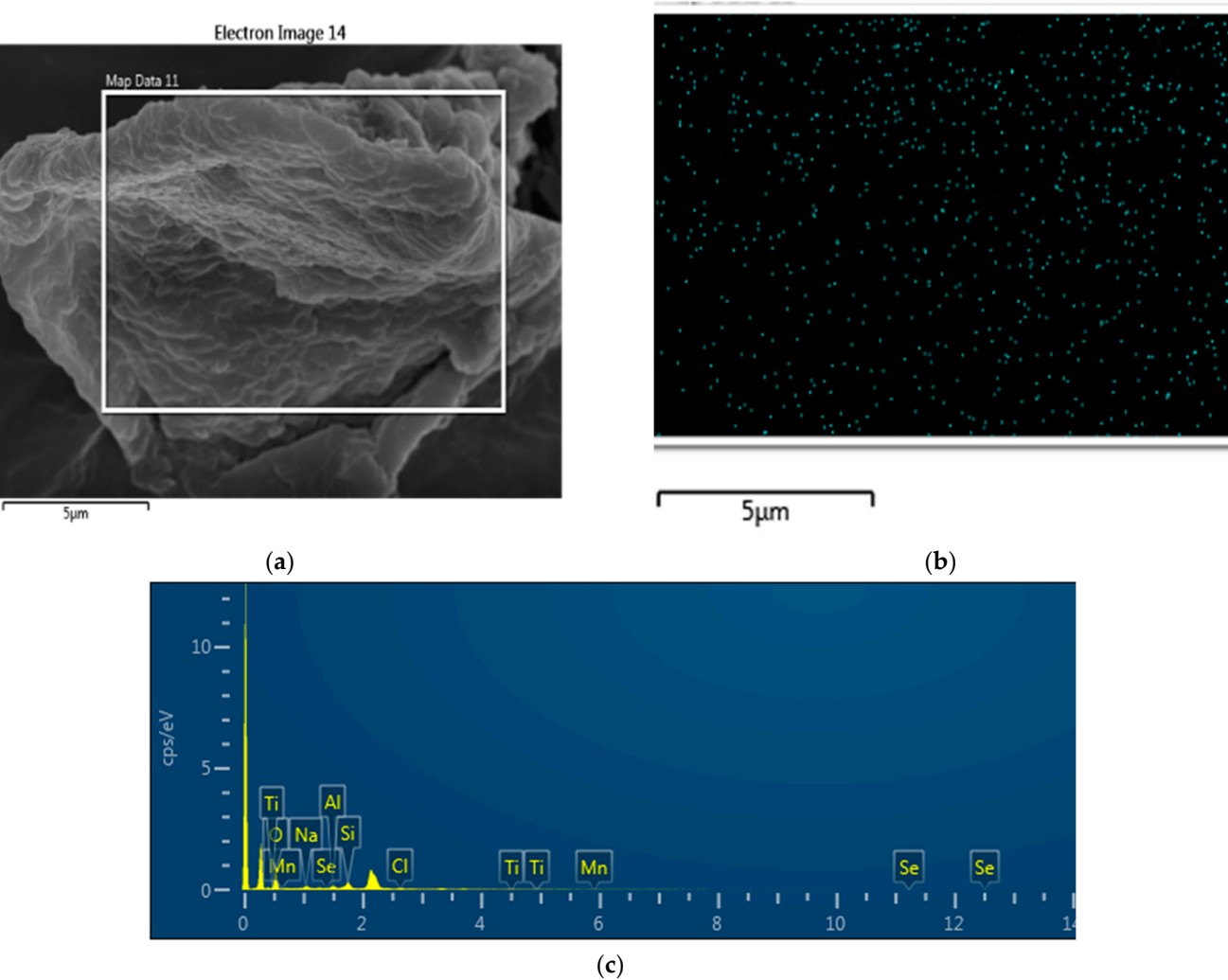

**Figure 3.** SEM–EDS analysis image of TMS clay rock: (**a**) SEM image of TMS clay particle; (**b**) EDS mapping analysis; (**c**) EDS spectra.

**Table 4.** Diffusion parameters of HTO and Se in TMS clay rocks in through-diffusion experiments.

| Item | HTO | | | | | Se |
|---|---|---|---|---|---|---|
| | No. 1 (1.6) | No. 2 (1.8) | No. 3 (2.0) | No. 4 (2.2) | No. 5 (2.4) | No. 6 (-) |
| $\alpha$ | 0.4074 | 0.3330 | 0.2593 | 0.1852 | 0.1111 | 2.9546 |
| $D_a \times 10^{-10}$ (m$^2$/s) | 1.92 | 1.77 | 2.49 | 2.84 | 3.53 | 0.011 |
| $D_e \times 10^{-11}$ (m$^2$/s) | 7.81 | 5.89 | 6.47 | 5.25 | 3.92 | 0.324 |
| $K_d$ (mL/g) | 0.00 | 0.00 | 0.00 | 0.00 | 0.00 | 1.10 |
| $t_d$ | 16.67 | 15.29 | 21.55 | 24.51 | 40.50 | 1.34 |
| R-squared | 0.99 | 0.99 | 0.99 | 0.99 | 0.99 | 0.97 |

For HTO: $\alpha = \theta$; Se: $\alpha = \theta + \rho_b \times K_d$.

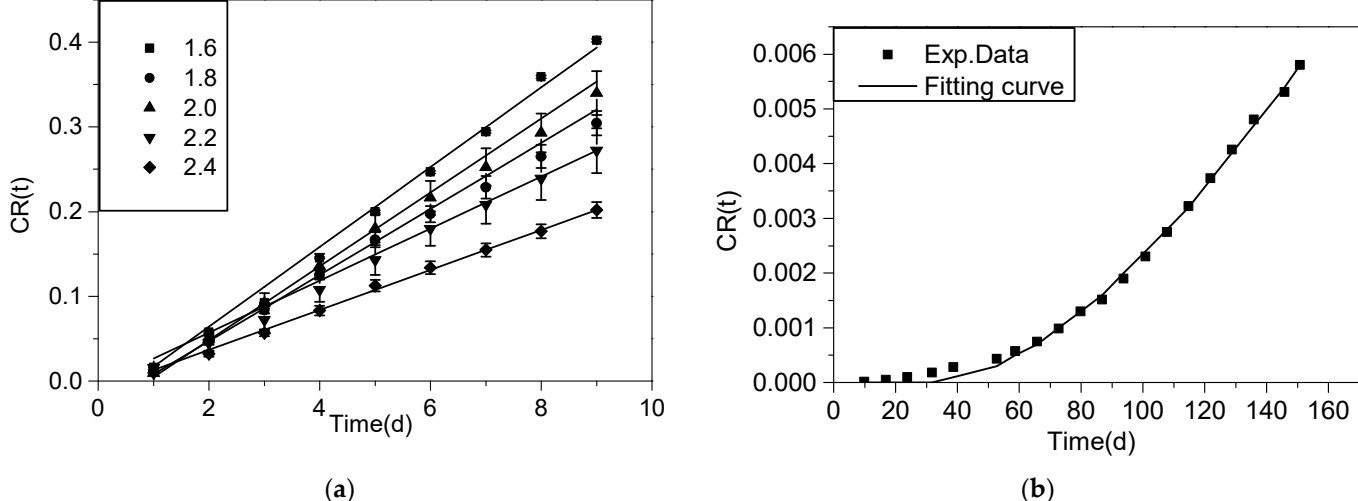

**Figure 4.** Diffusion of HTO and Se in compacted and intact TMS clay rocks: (**a**) HTO; (**b**) Se.

*3.4. Spatial and Temporal Variation with Various Diffusion Coefficients by Numerical Analysis*

According to the TD experimental results of HTO and Se in Figure 4 and Table 4, the diffusion coefficients ranged widely, from $10^{-10}$ to $10^{-12}$ m$^2$/s, and it required more time to reach steady-state diffusion in the Se(IV) TD experiments. In this study, a numerical analysis was applied to assess the spatial and temporal variation of $C(x.t)$ for the various diffusion coefficients by using Equation (3). Figure 5 and Table 5 show the spatial and temporal variation $C(x,t)$ of $D_a = 1.00 \times 10^{-10}, 1.00 \times 10^{-11}$, and $1.00 \times 10^{-12}$ m$^2$/s through a 0.3 cm thick sample and unit cross-sectional area at different times. However, there are also some limitations to TD experiments, such as detection uncertainty or errors. The numerical analysis of spatial variation for the concentration profiles depended on the diffusion flux ($C/C_0 > 0.001$) at various distances ($x$), set from 0.295 to 0.299 cm, and is given by Equation (3). According to the numerical analysis of spatial and temporal variation $C(x,t)$, shown in Figure 5a,b, at $D_a$ value = $1.00 \times 10^{-10}$ m$^2$/s, the straight line reveals a certain time at which the steady-state conditions reached were entirely different, ranging from 0.25 to 0.46 days (6 to 11 h). Moreover, it shows results (steady state) in agreement with the TD experimental results of HTO in compacted TMS clay rocks after 12 h. In contrast, a longer time, from 18 to 46.8 days, is shown in Figure 5e,f for the spatial variation $C(x,t)$ with a decreasing $D_a$ value, as determined by the Se(IV) TD experiments.

**Table 5.** Steady-state time in temporal and spatial variation with different $D_a$ values.

| $D_a$ (m$^2$/s) | T (Days) in Temporal Variation | T (Days) in Spatial Variation (x = 0.299 cm) |
|---|---|---|
| $1.00 \times 10^{-10}$ | 0.25 | 0.46 |
| $1.00 \times 10^{-11}$ | 2.25 | 4.70 |
| $1.00 \times 10^{-12}$ | 18.0 | 46.8 |

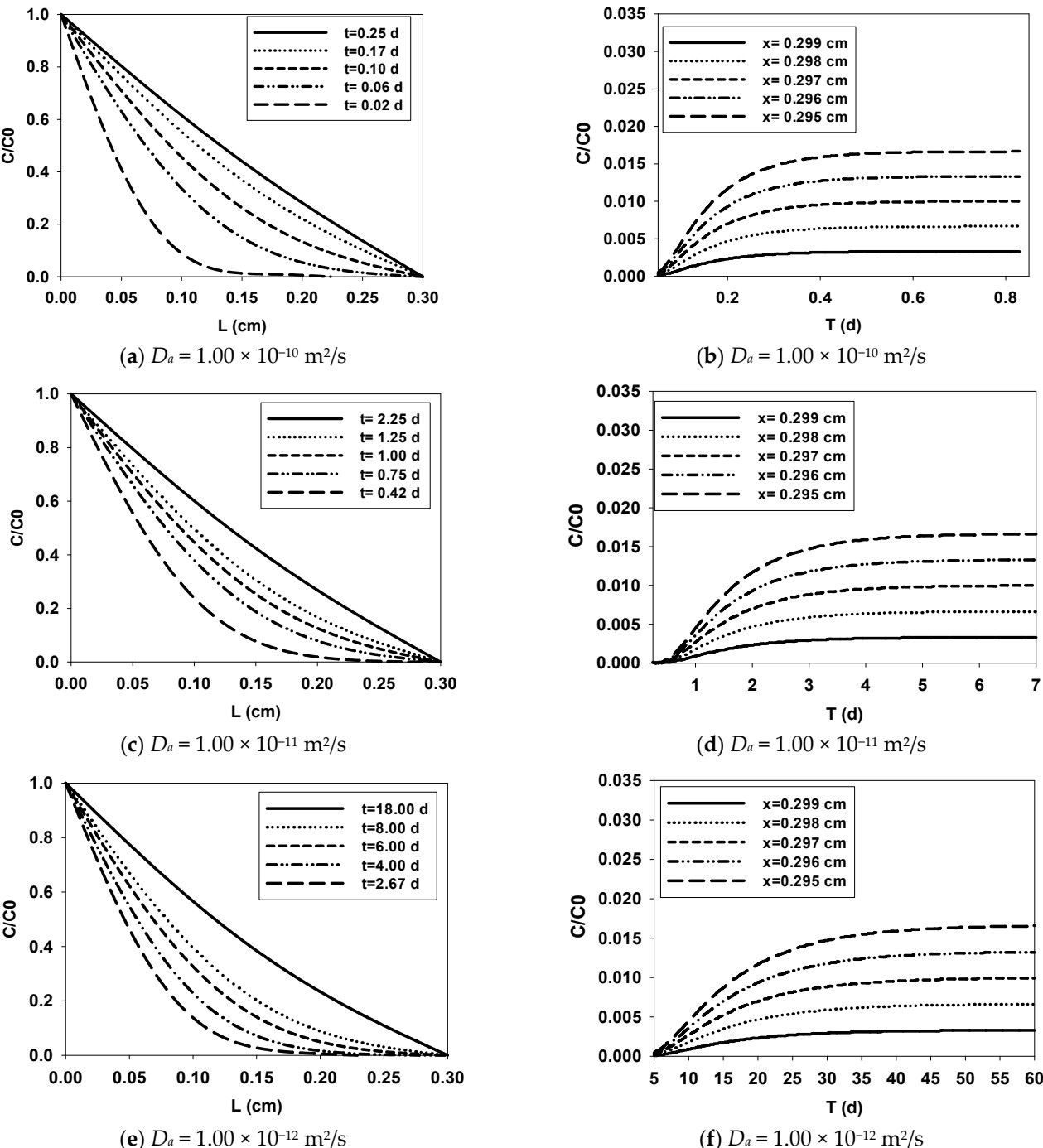

**Figure 5.** Numerical analysis of concentration profiles $C(x,t)$: (**a**) temporal: $C(x,t)$ at $D_a = 1 \times 10^{-10}$ m²/s and (**b**) spatial: $C(x,t)$ at $D_a = 1 \times 10^{-10}$ m²/s; (**c**) temporal: $C(x,t)$ at $D_a = 1 \times 10^{-11}$ m²/s and (**d**) spatial: $C(x,t)$ at $D_a = 1 \times 10^{-11}$ m²/s; (**e**) temporal: $C(x,t)$ at $D_a = 1 \times 10^{-12}$ m²/s and (**f**) spatial: $C(x,t)$ at $D_a = 1 \times 10^{-12}$ m²/s.

## 4. Discussion

The two-site sorption models in the batch sorption tests, which correspond to two decay constants, could be more adequate than the one-site models in realizing and characterizing the Se(IV) sorption mechanism. The batch sorption kinetic experiments indicated and suggested that sorption capacity plays an important role, affecting Se(IV) sorption on TMS clay rocks in this work. Through model calibration and validation ($R^2 > 0.9$), a set of parameters for the maximum capacity ($1.75 \times 10^{-4}$ mol/g) was determined, with which the batch experimental results of Se(IV) in GW could be described adequately. Compared

with TMS clay, responsible for the sorption of Se, the study also showed that clay minerals contain iron as a major component, and several studies have also reported different material analyses [27–30]. We found that Se(IV) has higher retardation than HTO by comparison between HTO and Se(IV) diffusion coefficients for TMS clay rock. In addition to the microporous composition (i.e., porosity), we also recognized that the key sorption (or retardation) of Se(IV) in TMS clay rock depended on the clay mineral composition (iron content), in agreement with the batch sorption and SEM–EDS experiments. For assessing spatial and temporal variability with various diffusion coefficients by numerical analysis, a good method would be to calculate and realize TD experiments by applying different $D_a$ values in the spatial and temporal variation concentration profile $C(x,t)$.

## 5. Conclusions

The diffusion behavior of Se(IV) and its characterization and sorption properties were investigated using batch sorption and column TD experiments. According to the coordinated experimental and numerical results of HTO and Se(IV), the diffusion of Se(IV) in TMS clay rocks was lower than that of HTO due to the clay mineral composition, and the TD experimental and numerical results were also numerically evaluated and demonstrated in TD tests. Numerical analysis of spatial and temporal variation could be a more effective and important tool in future safety assessments for clay rock repositories. Thus, the experimental and numerical results of HTO and Se(IV) in this work could be an important reference case for future safety assessments in clay rock repositories in China.

**Author Contributions:** Conceptualization, Y.S. (Yunfeng Shi) and N.-C.T.; Data curation, C.-P.L., J.K., Y.S. (Yunfeng Shi), S.-C.T., W.L. and Y.S. (Yuzhen Sun); Formal analysis, C.-P.L., J.K., Y.H., Y.S. (Yunfeng Shi) and N.-C.T.; Funding acquisition, Y.S. (Yunfeng Shi) and N.-C.T.; Investigation, C.-P.L.; Methodology, C.-P.L.; Project administration, N.-C.T. and S.-C.T.; Software, J.K. All authors have read and agreed to the published version of the manuscript.

**Funding:** This project was majorly supported by Doctor Initial Financial Project (No. 1410000434), East China University of Technology. This project was financed in part by the Ministry of Science and Technology (MOST, Taiwan R.O.C.) and the Atomic Energy Council (AEC, Taiwan R.O.C.) through a 2-year mutual fund program project under contract numbers 109-2623-E-007-006-NU and 110-2623-E-007-004-NU.

**Institutional Review Board Statement:** Not applicable.

**Informed Consent Statement:** Not applicable.

**Data Availability Statement:** Not applicable.

**Conflicts of Interest:** The authors declare no conflict of interest.

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
