# Peer review of "Molecule Diffusion Behavior of Tritium and Selenium in Mongolia Clay Rock by Numerical Analysis of the Spatial and Temporal Variation"

_minerals, doi:10.3390/min11080875_

Round 1

Reviewer 1 Report

In the presented work, the apparent diffusion coefficients of tritiated water HTO and selenium Se(IV) in compacted, finely ground TMS clay as well as in an "intact" but conditioned borehole clay rock sample of the same origin have been determined.

The conditioning of the intact probe included washing, rinsing, desiccating for one day and storing the probe for an unspecified time. Therefore, the diffusion coefficients of the laboratory sample and in actual in-situ rock may differ to some extent.

The diffusion coefficients of HTO and Se in the clay samples were obtained by modelling the concentration profiles with analytical expressions for an isotropic medium and using least square analysis to determine the optimal value of the apparent diffusion coefficients. Uncertainties on the results have not been not given.

Retardation was observed in the transport of Se. One-site and a two-site sorption models were tested and compared with experimental results. The two-site model was found to be suitable to explain the retardation of Se in intact TMS clay.

As far as the method, the experimental procedure, the mathematical treatment and the findings of the presented work are concerned, the manuscript looks sound to me. I merely find it regrettable that (i) no uncertainties on D values are given, (ii) the obtained results are not discussed in terms of their meaning for a (future) safety assessment, especially how the laboratory results are expected to relate to diffusion properties of in-situ rock.

The problem I have with the manuscript is that there are quite a lot of substantial imperfections in the language: Every third or fourth sentence I had to stop and think about what exactly the authors intended to say, guessing my way through the paper. The text needs a major revision. I am afraid that some readers will abandon studying the paper if this is not improved. 

In conclusion, I find the paper acceptable in principle, but I recommend that a great effort is put into reformulating almost the entire paper more intelligibly, preferably with the help of a colleague with a good command of English as well as a good understanding of the author's work. 

Author Response

Dear Reviewer

Thanks for your valuable comments and suggestion. Firstly, we have submitted to English editing by MDPI recommendation (https://www.mdpi.com/authors/english). For uncertainties in our diffusion coefficients have not been not showed, it could be explained that we applied numerical analysis by evaluating the concentration profiles with the lowest least-square value between experimental results and simulation value. Moreover, a set of 5 columns is compacted with decreasing bulk density (No. 1 ~No. 5), and effective diffusion coefficients of HTO also showed a tendency to decrease in Table 4. In fact, we applied TD columns for several years, and uncertainty could be estimated around 5 to 10% in our TD system according to our experience. Finally, we appreciate your kind and excellent suggestion for our manuscript, and Thanks a lot!

Best Regards

李传斌 博士(Chuan-Pin Lee Ph.D)

东华理工大学核科学与工程学院(School of Nuclear Science and Engineering, East China

University of Technology )

EMAIL:bennis6723@139.cn ; bennis6723@gmail.com

地址: 330013 中国江西省南昌市昌北经济技术开发区广兰大道418号

Reviewer 2 Report

Manuscript Number:  minerals-1276533

Authors:  Chuan-Pin Lee, Yanqin Hu, Yunfeng Shi, Neng-Chuan Tien, Shih-Chin Tsai, Jie Kong, Weigang Liu and Yuzhen Sun

Entitled: Molecule diffusion behavior of Tritium and Selenium in Mongolia clay rock by numerical analysis of the spatial and temporal variability

Journal: Minerals

General comments

The paper is very interesting and presents significant research in the field of the deep geological disposal of radioactive waste. I have no serious objections except one suggestion regarding the half-life of Se-79.

  1. Half life for Se-79 is 3.56 (40) *10^5 a

I recommend publication of this paper in the present form.

Author Response

Dear Reviewer

Thanks for your valuable comments.

We have correct Se-79 half-life in our manuscript, and Thanks a lot!

Best Regards

李传斌 博士(Chuan-Pin Lee Ph.D)

东华理工大学核科学与工程学院(School of Nuclear Science and Engineering, East China

University of Technology )

EMAIL:bennis6723@139.cn ; bennis6723@gmail.com

地址: 330013 中国江西省南昌市昌北经济技术开发区广兰大道418号

Round 2

Reviewer 1 Report

The authors have put a great effort in improving the writing of their manuscript. The text is now very readable and suitable for publication.

I still have a few minor points for consideration, but they are facultative:

I think there is a misconception in the manuscript regarding the use of the term "variability". If I am not mistaken, the authors, whenever they write "variability", they actually mean "variation". Variation means an amount of change of a measurable quantity in time (or space, etc...), e.g. of C(x) with t, or C(x,t). Variability means that some measurable quantity is different from one case to the other, e.g. the diffusion coefficient is different from sample to sample.

Here some more remarks:

Page 1 (Abstract)

"A numerical analysis with a minimum error for the HTO and Se(IV) diffusion coefficients in compacted TMS clay was conducted."

I suggest: "A minimum error analysis was conducted to determine the HTO and Se(IV) diffusion coefficients in compacted TMS clay."

Page 2

"can result in the acceleration of radionuclide migration" -> do you mean: "can result in faster radionuclide migration" ? (acceleration is the change of the change of something).

I still don't understand this sentence:

"Furthermore, in a groundwater the high adsorption for cations strengthens simultaneously the anion exclusion effect between negative radionuclides and the permanent negative charge on the clay surface."

Page 3

"where the reservoir containing the tracer has a constant concentration (C0), the measurement reservoir is kept close to zero."

I suggest: "where the concentration in the reservoir containing the tracer is constant (C0) and the concentration in the opposing reservoir is kept close to zero." 

Page 11

"of Da = 1.00 × 10−10, 1.00 × 10−1, and 1.00 × 10−12 m2/s"

A digit got lost in the second exponent, wich should be -11.

Page 12

Something is missing in this sentence:

"Compared with TMS clay, responsible for the sorption of Se, the study also showed that clay minerals containing iron as a major component, and several studies have also reported different material analyses."

Author Response

Dear Reviewer

We are very grateful to you for all your kind assistance, especially for our English and professional comments.

Several terms and sentences have been modified according to your suggestion. Thanks again.

Best Regards

NC Tien

CP Lee